# Platelet-Rich Plasma in Chronic Wound Management: A Systematic Review and Meta-Analysis of Randomized Clinical Trials

**DOI:** 10.3390/jcm11247532

**Published:** 2022-12-19

**Authors:** Fanni Adél Meznerics, Péter Fehérvári, Fanni Dembrovszky, Kata Dorottya Kovács, Lajos Vince Kemény, Dezső Csupor, Péter Hegyi, András Bánvölgyi

**Affiliations:** 1Department of Dermatology, Venereology and Dermatooncology, Faculty of Medicine, Semmelweis University, 1085 Budapest, Hungary; 2Centre for Translational Medicine, Semmelweis University, 1085 Budapest, Hungary; 3Department of Biostatistics, University of Veterinary Medicine, 1085 Budapest, Hungary; 4Institute for Translational Medicine, Medical School, Szentágothai Research Centre, University of Pécs, 7624 Pécs, Hungary; 5HCEMM-SU Translational Dermatology Research Group, Department of Physiology, Semmelweis University, 1094 Budapest, Hungary; 6Institute of Clinical Pharmacy, Faculty of Pharmacy, University of Szeged, 6720 Szeged, Hungary; 7Division of Pancreatic Diseases, Heart and Vascular Center, Semmelweis University, 1085 Budapest, Hungary

**Keywords:** wound healing, dressing, platelet-rich plasma

## Abstract

Background: Chronic wounds place a heavy burden on the healthcare system due to the prolonged, continuous need for human resources for wound management. Our aim was to investigate the therapeutic effects of platelet-rich plasma on the treatment of chronic wounds. Methods: The systematic literature search was performed in four databases. Randomized clinical trials reporting on patients with chronic wounds treated with platelet-rich plasma (PRP) were included, comparing PRP with conventional ulcer therapy. We pooled the data using the random effects model. Our primary outcome was the change in wound size. Results: Our systematic search provided 2688 articles, and we identified 48 eligible studies after the selection and citation search. Thirty-three study groups of 29 RCTs with a total of 2198 wounds showed that the odds for complete closure were significantly higher in the PRP group than in the control group (OR = 5.32; CI: 3.37; 8.40; I^2^ = 58%). Conclusions: PRP is a safe and effective modality to enhance wound healing. By implementing it in clinical practice, platelet-rich plasma could become a widely used, valuable tool as it could not only improve patients’ quality of life but also decrease the healthcare burden of wound management.

## 1. Introduction

Chronic wounds are common conditions that greatly impact patients’ quality of life [1]. They place a heavy burden on the healthcare system due to the high cost of dressing materials, amputation-related costs, and the prolonged, continuous need for human resources for wound management [2].

The wide range of causes underlying ulceration includes arterial and venous insufficiency, neuropathy, microangiopathy, and several additional factors [3]. Besides treating the underlying cause, the goal of ulcer management is to promote healing through professional wound care; the gold standard methods are smart dressings and compression therapy [4].

Platelet-rich plasma (PRP) is an autologous serum prepared from whole blood by centrifugation, containing high concentrations of platelets, growth factors, and cytokines, which can promote stem cell regeneration and tissue remodeling [5,6]. By potentially shortening the recovery time of ulcers, PRP, as an additional treatment modality, could improve patients’ quality of life and decrease the healthcare burden of wound management.

Although the effects of PRP on wound healing are heavily investigated, the current evidence is inconclusive [7]. Our goal is to investigate the therapeutic effect of PRP on the treatment of chronic wounds by summarizing the latest data in a comprehensive manner by conducting a systematic review and meta-analysis.

## 2. Materials and Methods

Our study was performed according to the Cochrane Handbook’s recommendations for the Systematic Reviews of Interventions, Version 6.3 [8]. The results are reported following the guidelines of the PRISMA (Preferred Reporting Items for Systematic Reviews and Meta-Analyses) 2020 Statement [9]. The review protocol was registered on PROSPERO under registration number CRD42021287881 (see https://www.crd.york.ac.uk/prospero, accessed on 28 October 2021); no amendments to the information provided at registration were made.

The systematic literature search was performed in four databases: MEDLINE (via PubMed), Cochrane Library (CENTRAL), Embase, and Web of Science from inception to 29 October 2021. The query (ulcer * OR chronic ulcer OR chronic wound OR diabetic foot) AND (platelet rich plasma OR PRP OR platelet rich plasma gel OR PRPG OR platelet rich in growth factors OR PRGF) was applied to all fields in the search engines. No language or other restrictions were imposed.

Randomized clinical trials (RCTs) reporting on patients with chronic wounds treated with PRP were included, comparing additional PRP treatment with conventional ulcer therapy alone. The following population–intervention–control–outcome (PICO) framework was used:P—Adult patients with chronic wounds;I—Platelet-rich plasma (PRP) treatment;C—Conventional ulcer therapy;O—Primary outcome: change in wound size (complete closure, reduction of wound area, healing rate); secondary outcomes: healing time, infection, pain, adverse events, amputation, recurrence, and quality of life.

EndNote X9 (Clarivate Analytics, Philadelphia, PA, USA) was used for the selection of the articles. Two independent authors (F.A.M. and K.D.K.) screened the publications separately for the title, abstract (Cohen’s Kappa: 0.81), and full text (Cohen’s Kappa: 0.88), and disagreements were resolved by a third author (F.D.).

Two authors (F.A.M. and K.D.K.) independently extracted the data into an Excel spreadsheet (Office 365, Microsoft, Redmond, WA, USA). We collected the following data from the eligible articles: first author, year of publication, study type, study location, number of centers included in the study, study design, demographic data, details of the received treatments, and data regarding our outcomes for statistical analysis. A third reviewer (F.D.) resolved the discrepancies. Secondary outcomes were included if three publications reporting on them were found.

The quality assessment of the outcomes was performed separately by two reviewers (F.A.M. and K.D.K.) using the revised tool for assessing the risk of bias (RoB 2) [10]. A third reviewer (F.D.) resolved any occurring disagreements. To assess the quality of the evidence, we followed the recommendation of the “Grades of Recommendation, Assessment, Development, and Evaluation (GRADE)” workgroup [11].

The statistical analyses were made with R (R Core Team 2022, v4.2.1) [12]. For calculations and plots, we used the meta (Schwarzer 2022, v5.5.0) [13] and dmetar (Cuijpers, Furukawa, and Ebert 2022, v0.0.9000) [14] packages.

For the dichotomous outcomes, the odds ratio (OR) with a 95% confidence interval (CI) was used for the effect measure; to calculate the OR, the total number of patients in each group and those with the event of interest were extracted from each study. Raw data from the selected studies were pooled using a random effect model with the Mantel-Haenszel method [15,16,17]. For the pooled results, the exact Mantel–Haenszel method (no continuity correction) was used to handle zero cell counts [18]. In individual studies, the zero cell count problem was adjusted by treatment arm continuity correction [19]. In the case of continuous outcomes, a standardized mean difference (SMD) with a 95% CI was calculated as the effect size. As different results were used from the same study, a three-level meta-analysis model was used along with estimating an additional within the study heterogeneity variance parameter. The inverse variance weighting method was used to calculate the pooled SMD. To estimate the heterogeneity variance measure, τ^2^, the restricted maximum-likelihood estimator was applied with a t-distribution-based confidence interval [20].

Between-study heterogeneity was described by Higgins and Thompson’s I^2^ statistics [21]. As the subgroup analysis, the fixed-effects (plural) model (aka. the mixed-effects model) was used. Common τ values at the subgroup levels were assumed in the subgroup analysis, as we had a limited number of studies in some groups. A “Q” omnibus test (of all levels of the subgroup) was also calculated for comparison of the subgroup’s pooled effect sizes. If the study number for the given outcome was over five, the Hartung–Knapp adjustment [22] was applied (below six studies, no adjustment was applied).

A funnel plot of the logarithm of the effect size and comparison with the standard error for each trial was used to evaluate publication bias. Publication bias was assessed with Egger’s test using the Harbord method [23] to calculate the test statistic. Outlier and influence analyses were carried out following the recommendations of Harrer et al. [20] and Viechtbauer and Cheung [24].

## 3. Results

Our systematic search provided a total of 2688 articles; after duplicate removal, we screened 1910 duplicate-free articles. Following the title, abstract, and full-text selection, we identified 46 RCTs matching our PICO framework [25,26,27,28,29,30,31,32,33,34,35,36,37,38,39,40,41,42,43,44,45,46,47,48,49,50,51,52,53,54,55,56,57,58,59,60,61,62,63,64,65,66,67,68,69,70] and two additional articles [71,72] after the citation search. The full text of 10 articles could not be retrieved, even after contacting the authors [73,74,75,76,77,78,79,80,81,82]. The summary of the selection process is shown in Figure 1.

We conducted a quantitative analysis of our primary outcome, the change in wound size. The secondary outcomes are detailed in the systematic review section due to the widely varying and poorly defined outcome measures used for their assessment.

The characteristics of the identified RCTs for the systematic review and meta-analysis are detailed in Table 1.

### 3.1. Primary Outcome

The results of the studies assessing the change in wound size are detailed in Appendix A. Studies evaluating the change in wound size by measuring the baseline and post-treatment wound size or complete closure are included in our quantitative analysis.

#### 3.1.1. Complete Closure

Thirty-three study groups of 29 RCTs with a total of 2198 wounds showed that the odds for complete closure were significantly higher in the PRP group than in the control group (OR = 5.32; CI: 3.37; 8.40; I^2^ = 58%) (see Figure 2).

When subgrouping was based on ulcer etiologies, the odds for complete closure were significantly higher in the PRP group than in the control group, both in diabetic foot ulcers (OR = 2.26; CI: 1.50; 3.41; I^2^ = 12.0%) as well as venous leg ulcers (OR = 8.02; CI: 3.63; 17.71; I^2^ = 10.0%). The test for the subgroup difference showed a significant difference between the two groups (χ^2^ = 9.88; df = 1; *p* = 0.002); the odds for complete closure were significantly higher in venous ulcers than in the diabetic foot ulcers treated with PRP (see Appendix A).

Subgrouping based on the way PRP was applied showed similar results. The odds for complete closure were significantly higher both in the topically applied (OR = 4.74; CI: 2.87; 7.83; I^2^ = 60%) and injected (OR = 9.42; CI: 3.32; 26.76; I^2^ = 0%) PRP groups than in the control group, with no significant subgroup difference (χ^2^ = 2.34; df = 1; *p* = 0.126) (see Appendix A).

The odds for complete closure were significantly higher in the PRP group than in the control group in the short (OR = 6.03; CI: 3.21; 11.33; I^2^ = 47%), medium (OR = 3.38; CI: 1.15; 9.89; I^2^ = 73%), and long (OR = 8.24; CI: 1.66; 40.87; I^2^ = 0%) follow-up categories, as well with no significant subgroup differences (χ^2^ = 2.50; df = 3; *p* = 0.476) (see Appendix A).

#### 3.1.2. Reduction of Wound Area

The pooled SMDs from 18 study groups of 16 RCTs with a total of 1062 wounds showed a significant difference between the post-treatment wound size of the PRP and the control groups (SMD = −1.21, CI: −1.74; −0.68; I^2^ = 92.5%), with the PRP group showing greater improvement (see Figure 3).

Subgrouping based on ulcer etiology, the application method, and follow-up length showed similar results (see Appendix A). The post-treatment wound size was significantly smaller in the PRP group than in the control group in the diabetic (SMD = −0.68, CI: −1.31; −0.06; I^2^ = 93.64%), venous (SMD = −1.26, CI: −2.28; −0.24; I^2^ = 90.76%), topically applied (SMD = −0.94, CI: −1.43; −0.46; I^2^ = 91.26%), and injected (SMD = −1.03, CI: −1.79; −0.26; I^2^ = 86.63%) subgroups, as well as in the short follow-up subgroup (SMD = −1.00, CI: −1.64; −0.35; I^2^ = 89.41%). However, the difference between the PRP and the control groups was not significant in the medium (SMD = −1.38, CI: −2.96; 0.19; I^2^ = 54.51%) and long (SMD = −0.63, CI: −1.64; 0.37; I^2^ = 93.88%) follow-up groups. No significant subgroup differences were recorded.

### 3.2. Secondary Outcomes

The secondary outcomes are summarized in Table 2. Recurrence rates and quality of life are not reported, as less than three studies included them as an outcome.

### 3.3. Risk of Bias Assessment

The result of the assessment of the risk of bias of the studies included in the meta-analysis and systematic review are detailed in Appendix A. None of the studies included in the meta-analysis was at a high risk of bias. In thirty studies [26,28,29,30,32,34,39,42,43,47,48,49,50,51,52,53,56,58,59,61,62,64,65,66,67,68,69,70,71,72], the ‘randomization process’ domain, in twelve studies [28,42,44,48,50,53,56,59,63,65,68,71], the ‘deviations from intended interventions’ domain, in one study [29], the ‘missing outcome data’ domain, in five studies [44,50,56,59,71], the ‘measurement of the outcome’ domain, and in eight studies [26,33,42,57,58,63,65,70], the ‘selection of the reported result’ domain, were rated as ‘some concerns’ for our primary outcome.

### 3.4. Quality of Evidence

The quality of the evidence for our outcomes is detailed in the Summary of Findings Table (see Appendix A).

### 3.5. Publication Bias

The funnel plot assessing the publication bias can be seen in the Appendix A. No evidence of serious publication bias can be observed in the funnel plot for complete closure; however, the funnel plot for the reduction of the wound area indicates publication bias.

## 4. Discussion

On the basis of our systematic review and meta-analysis, PRP is an effective add-on treatment modality to enhance wound healing. The odds for complete wound closure were significantly higher in the PRP group than in the control group, and PRP also resulted in a significantly greater reduction of the wound area compared to conventional therapy.

The subgroup analyses, which were conducted to decrease the heterogeneity, showed similar results and also highlighted differences between the ulcer etiologies and PRP application methods. Injected PRP seemed to result in greater improvement than topically applied PRP; however, due to the relatively low sample size of this subgroup, conclusions should be drawn with caution. As for ulcer etiologies, while PRP was superior to conventional therapy regarding complete closure and the reduction of the wound area in diabetic and venous ulcers as well, better results were recorded in the venous ulcer group. The reason for this phenomenon could be that diabetic ulcers are more difficult to heal; however, the fact that PRP was more frequently administered by injection in the venous ulcer group could also be a contributing factor, as we saw better results in the injected PRP subgroup discussed above. PRP was also shown to be effective after short, medium, and long follow-up times regarding complete closure.

Although we did not conduct quantitative analysis on the healing time due to the varying reporting methods of the studies, all the included studies reported shorter healing times in the PRP group than in the conventional therapy group [27,34,35,36,37,38,43,47,48,52,53,59,63,69].

The infection rate is another critical outcome that requires further investigation with more specific criteria for its assessment. Nine studies did not record a significant difference between the PRP and the control groups regarding infection rates [29,30,33,38,46,48,57,58,72], whereas four studies recorded a significantly lower number of infections in the PRP group [25,26,28,69], suggesting that PRP could decrease the risk of infection.

No substantial difference was recorded between the PRP and the control group regarding pain [25,28,30,31,38,39,43,46,51,60,69], amputation rates [38,42,45,48,55,63], and adverse events [30,31,32,34,38,43,47,48,51,57,61,63,69,72].

### 4.1. Strengths and Limitations

There are several strengths to our study. We summarized the latest data on PRP in wound management in a comprehensive manner, assessing the most objective outcome measure, the change in the wound area. Our results clearly support the superiority of PRP over conventional therapy alone. While previous studies only assessed the efficacy of PRP in different ulcer etiologies separately, we conducted an overall analysis; we believe, as well, that it is crucial to assess the wound-healing properties of PRP in general [7]. We only included RCTs and implemented a rigorous methodology to guarantee the highest possible quality of evidence and conducted a quantitative analysis only on the outcomes that were objectively reported to avoid drawing false conclusions based on poorly recorded secondary outcomes. Our limitations included publication bias and the diversity of the control groups, as a wide range of dressings was used as a part of the conventional therapy.

### 4.2. Implications for Research

Future studies should report their outcomes uniformly to enable further comprehensive analysis. As the most objective way of assessing the clinical efficacy of PRP in wound management is to record the change in wound size, the baseline and post-treatment wound area should always be reported. However, better reporting guidelines are required that entail detailed descriptive statistics, including the median and interquartile range besides the mean and standard deviation. Additionally, the varying methods used to measure wound size can also lead to further bias: chronic wounds often affect the leg, and simply photographing the wound and measuring it with software does not take into account that wounds often affect the total leg circumference. Also, assessing the wound size by only measuring its width and length can give false results due to the often asymmetrical ulcer areas. We suggest that the most applicable way of precise measurement is tracing the outline of the wound on carbon paper, which can be digitalized and available for further calculations.

In addition to the baseline and post-treatment wound area, the number of completely closed wounds is also a critical outcome measure, showing the clinical efficacy of the treatment; therefore, it should always be reported.

### 4.3. Implications for Practice

The importance of the early application of research results in clinical practice is undisputable [83]. Due to its wound-healing properties, platelet-rich plasma could become a widely used, valuable tool in chronic wound management. PRP can be administered topically and intralesionally, as well, and can also be applied along with the wide range of available smart dressings. These combinations enable personalized treatment strategies by providing a variety of options for treating physicians.

## 5. Conclusions

Platelet-rich plasma is a safe and effective modality to enhance wound healing. By implementing it in clinical practice, PRP could become a widely used, valuable tool, as it could improve patients’ quality of life and decrease the healthcare burden of wound management.

## Figures and Tables

**Figure 1 jcm-11-07532-f001:**
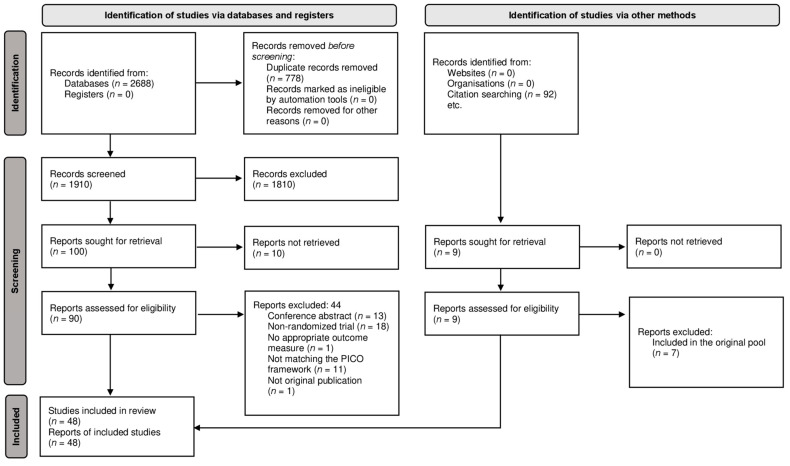
PRISMA Flow Diagram of the screening and selection process.

**Figure 2 jcm-11-07532-f002:**
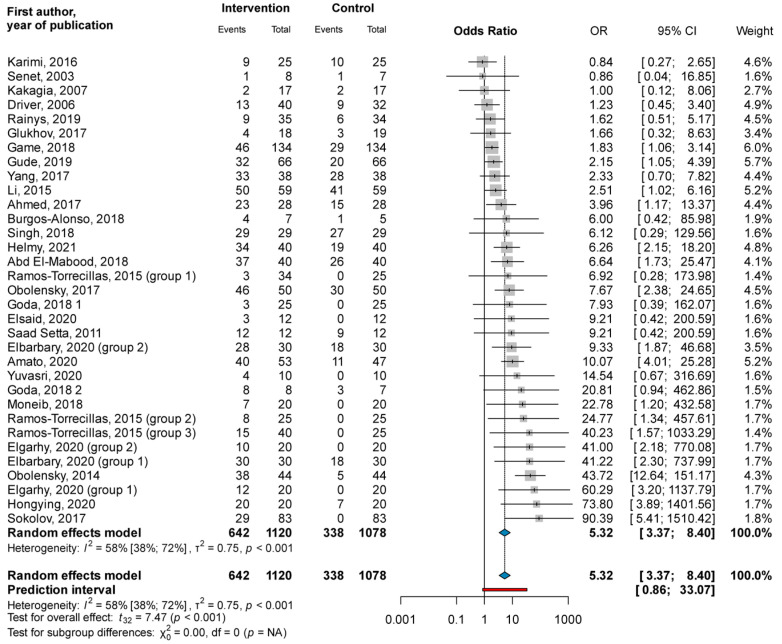
Forest plot for complete closure: platelet-rich plasma compared to conventional ulcer therapy [25,26,28,30,34,35,36,37,38,39,40,41,42,43,44,45,48,51,52,53,57,58,59,63,64,69,70,71,72].

**Figure 3 jcm-11-07532-f003:**
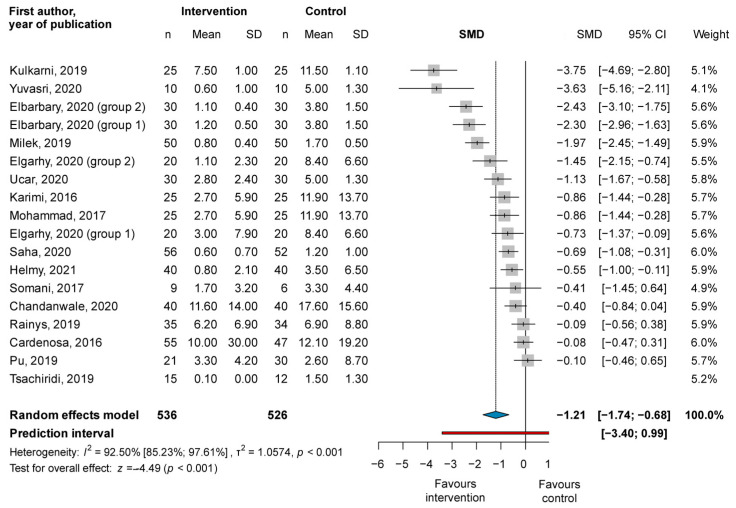
Forest plot for the change of wound size: platelet-rich plasma compared to conventional ulcer therapy [31,32,35,36,43,45,47,49,50,55,57,60,65,66,68,70].

**Table 1 jcm-11-07532-t001:** Characteristics of the included studies.

First Author, Year of Publication	Type of Publication	Study Type	Country	Ulcer Etiology	Outcome
Abd El-Mabood, 2018 [25]	Journal article	RCT	Egypt	Diabetic	Complete closure, healing rate, infection, and pain
Ahmed, 2017 [26]	Journal article	RCT	Egypt	Diabetic	Complete closure, healing rate, and infection
Alamdari, 2021 [27]	Journal article	RCT	Iran	Diabetic	Healing time, and amputation
Amato, 2020 [28]	Journal article	RCT	Italy	Mixed	Reduction of wound area, complete closure, infection, and pain
Anitua, 2008 [29]	Journal article	RCT	Spain	Mixed	Reduction of wound area and infection
Burgos-Alonso, 2018 [30]	Journal article	RCT	Spain	Venous	Reduction of wound area, complete closure, infection, pain, adverse events, and quality of life
Cardenosa, 2017 [31]	Journal article	RCT	Spain	Venous	Reduction of wound area, pain, and adverse events
Chandanwale, 2020 [32]	Journal article	RCT	India	Arterial	Reduction of wound area
de Oliveira, 2017 [33]	Journal article	RCT	Brazil	Venous	Reduction of wound area and infection
Driver, 2006 [34]	Journal article	RCT	US	Diabetic	Reduction of wound area, healing rate, complete closure, healing time, and adverse events
Elbarbary, 2020 [35]	Journal article	RCT	India	Venous	Reduction of wound area, complete closure, healing time, and recurrence
Elgarhy, 2020 [36]	Journal article	RCT	India	Venous	Reduction of wound area, complete closure, and healing time
Elsaid, 2020 [37]	Journal article	RCT	Egypt	Diabetic	Reduction of wound area, complete closure, and healing time
Game, 2018 [38]	Journal article	RCT	UK	Diabetic	Reduction of wound area, complete closure, healing time, infection, pain, amputation, and adverse events
Glukhov, 2017 [39]	Journal article	RCT	Russia	Venous	Complete closure, and pain
Goda, 2018 1 [41]	Journal article	RCT	Egypt	Diabetic	Healing rate, and complete closure
Goda, 2018 2 [40]	Journal article	RCT	Egypt	Venous	Reduction of wound area, and complete closure
Gude, 2019 [42]	Journal article	RCT	US	Diabetic	Complete closure, and amputation
Helmy, 2021 [43]	Journal article	RCT	Egypt	Venous	Reduction of wound area, complete closure, healing time, pain, adverse events, and recurrence
Hongying, 2020 [44]	Journal article	RCT	China	Pressure	Reduction of wound area, and complete closure
Kakagia, 2007 [71]	Journal article	RCT	Greece	Diabetic	Reduction of wound area, and complete closure
Karimi, 2016 [45]	Journal article	RCT	Iran	Diabetic	Reduction of wound area, complete closure, and amputation
Khorvash, 2017 [46]	Journal article	RCT	Iran	Diabetic	Reduction of wound area, infection, pain, and quality of life
Kulkarni, 2019 [47]	Journal article	RCT	India	N/A	Reduction of wound area, healing time, and adverse events
Li, 2015 [48]	Journal article	RCT	China	Diabetic	Reduction of wound area, complete closure, healing time, infection, amputation, and adverse events
Milek, 2019 [49]	Journal article	RCT	Poland	Venous	Reduction of wound area and complete closure
Mohammad, 2017 [50]	Journal article	RCT	Iran	Diabetic	Reduction of wound area
Moneib, 2018 [51]	Journal article	RCT	Egypt	Venous	Reduction of wound area, complete closure, pain, and adverse events
Obolenskiy, 2014 [53]	Journal article	RCT	Russia	Mixed	Complete closure and healing time
Obolenskiy, 2017 [52]	Journal article	RCT	Russia	Mixed	Healing rate, complete closure, and healing time
Pires, 2021 [54]	Journal article	RCT	Brazil	Venous	Infection
Pu, 2019 [55]	Journal article	RCT	China	Arterial	Reduction of wound area, healing rate, and amputation
Qin, 2019 [56]	Journal article	RCT	China	Diabetic	Reduction of wound area
Rainys, 2019 [57]	Journal article	RCT	Lithuania	N/A	Reduction of wound area, complete closure, infection, and adverse events
Ramos-Torrecilla, 2015 [58]	Journal article	RCT	Spain	Pressure	Reduction of wound area, complete closure, and infection
Saad Setta, 2011 [59]	Journal article	RCT	Egypt	Diabetic	Complete closure and healing time
Saha, 2020 [60]	Journal article	RCT	India	Leprosy	Reduction of wound area, complete closure, and pain
Semenic, 2018 [61]	Journal article	RCT	Slovenia	Mixed	Reduction of wound area and adverse events
Senet, 2003 [72]	Journal article	RCT	France	Venous	Reduction of wound area, healing rate, complete closure, infection, and adverse events
Singh, 2018 [63]	Journal article	RCT	India	Diabetic	Complete closure, healing time, amputation, and adverse events
Singh, 2021 [62]	Journal article	RCT	India	Pressure	Reduction of wound area
Sokolov, 2017 [64]	Journal article	RCT	Bulgaria	Not defined	Complete closure
Somani, 2017 [65]	Journal article	RCT	India	Venous	Reduction of wound area and complete closure
Tsachiridi, 2019 [66]	Journal article	RCT	Greece	Pressure	Reduction of wound area and healing rate
Tsai, 2019 [67]	Journal article	RCT	US	Mixed	Reduction of wound area
Ucar, 2020 [68]	Journal article	RCT	Turkey	Pressure	Reduction of wound area
Yang, 2017 [69]	Journal article	RCT	China	Diabetic	Healing rate, healing time, infection, pain, and adverse events
Yuvasri, 2020 [70]	Journal article	RCT	India	Venous	Reduction of wound area and complete closure

**Table 2 jcm-11-07532-t002:** Main conclusions of the studies assessing the secondary outcomes.

First Author, Year of Publication	Main Conclusion
Healing Time	
Alamdari, 2021 [27]	Shorter healing time in the PRP group than in the control group
Driver, 2006 [34]	Shorter healing time in the PRP group than in the control group
Elbarbary, 2020 [35]	Shorter healing time in the PRP group than in the control group *
Elgarhy, 2020 [36]	Shorter healing time in the topical and injected PRP groups than in the control group *
Elsaid, 2020 [37]	Shorter healing time in the PRP group than in the control group *
Game, 2018 [38]	Shorter healing time in the PRP group than in the control group *
Helmy, 2021 [43]	Shorter healing time in the PRP group than in the control group *
Kulkarni, 2019 [47]	Shorter healing time in the PRP group than in the control group *
Li, 2015 [48]	Shorter healing time in the PRP group than in the control group *
Obolenskiy, 2014 [53]	Shorter healing time in the PRP group than in the control group
Obolenskiy, 2017 [52]	Shorter healing time in the PRP group than in the control group *
Saad Setta, 2011 [59]	Shorter healing time in the PRP group than in the control group *
Singh, 2018 [63]	Shorter healing time in the PRP group than in the control group *
Yang, 2017 [69]	Shorter healing time in the PRP group than in the control group *
Infection Rates	
Abd El-Mabood, 2018 [25]	More infection in the control group than in the PRP group *
Ahmed, 2017 [26]	More infection in the control group than in the PRP group *
Amato, 2020 [28]	More infection in the control group than in the PRP group *
Anitua, 2008 [29]	No statistically significant difference between the PRP and the control groups
Burgos-Alonso, 2018 [30]	No statistically significant difference between the PRP and the control groups
de Oliveira, 2017 [33]	No statistically significant difference between the PRP and the control groups
Game, 2018 [38]	No statistically significant difference between the PRP and the control groups
Khorvash, 2017 [46]	No statistically significant difference between the PRP and the control groups
Li, 2015 [48]	No statistically significant difference between the PRP and the control groups
Pires, 2021 [54]	No statistically significant differences in antimicrobial resistance between *P. aeruginosa* and *S. aureus* in the PRP and control groups. PRP decreased bacteriological growth or the microbial load and resistance profile in the case of *P. aeruginosa*
Rainys, 2019 [57]	No statistically significant difference between the PRP and the control groups
Ramos-Torrecilla, 2015 [58]	No signs of infection were recorded during the study
Senet, 2003 [72]	No statistically significant difference between the PRP and the control groups
Yang, 2017 [69]	More infection in the control group than in the PRP group *
Pain	
Abd El-Mabood, 2018 [25]	Pain occurred more frequently in the control group *
Amato, 2020 [28]	Pain occurred more frequently in the control group *
Burgos-Alonso, 2018 [30]	No statistically significant difference in pain reduction between the PRP and the control groups
Cardenosa, 2017 [31]	Pain reduction was higher in the PRP group *
Game, 2018 [38]	No statistically significant difference in pain reduction between the PRP and the control groups
Glukhov, 2017 [39]	All patients subjectively experienced pain reduction in both groups
Helmy, 2021 [43]	All patients subjectively experienced pain reduction in the PRP group
Khorvash, 2017 [46]	pain reduction was higher in the PRP group *
Moneib, 2018 [51]	All patients subjectively experienced pain reduction in both groups
Saha, 2020 [60]	Administration-related pain was reported by 10 participants in the PRP group
Yang, 2017 [69]	pain reduction was higher in the PRP group *
Amputation Rates	
Alamdari, 2021 [27]	No statistically significant difference between the PRP and the control groups
Game, 2018 [38]	No statistically significant difference between the PRP and the control group
Gude, 2019 [42]	Two amputations in the control group and no amputation in the PRP group
Karimi, 2016 [45]	No statistically significant difference between the PRP and the control groups
Li, 2015 [48]	Four amputations in the control group one amputation in the PRP group
Pu, 2019 [55]	No statistically significant difference between the PRP and the control groups
Singh, 2018 [63]	Two amputations in the control group, and no amputation in the PRP group
Adverse Events	
Burgos-Alonso, 2018 [30]	No statistically significant difference between the PRP and the control groups
Cardenosa, 2017 [31]	No adverse events recorded
Chandanwale, 2020 [32]	No adverse event in the PRP group
Driver, 2006 [34]	No administration related serious adverse event was recorded in either group; one case of Contact dermatitis in the PRP group and one case of maceration in the control group
Game, 2018 [38]	No statistically significant difference between the PRP and the control groups
Helmy, 2021 [43]	No adverse events recorded
Kulkarni, 2019 [47]	No adverse event in the PRP group
Li, 2015 [48]	No adverse events were recorded in the PRP group
Moneib, 2018 [51]	No adverse events recorded
Rainys, 2019 [57]	No statistically significant difference between the PRP and the control groups, and no serious adverse event was recorded
Semenic, 2018 [61]	No adverse events recorded
Senet, 2003 [72]	No statistically significant difference between the PRP and the control groups
Singh, 2018 [63]	No adverse events recorded
Yang, 2017 [69]	No statistically significant difference between the PRP and the control groups

PRP-platelet-rich plasma; * indicates significant difference (*p* < 0.05).

## Data Availability

The datasets used in this study can be found in the full-text articles included in the systematic review and meta-analysis.

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
