# Peer review of "Platelet-Rich Plasma in Chronic Wound Management: A Systematic Review and Meta-Analysis of Randomized Clinical Trials"

_jcm, 2022, doi:10.3390/jcm11247532_

Round 1
Reviewer 1 Report
For this review article, the authors have selected a topic with high potential. Manuscript is very well written covering almost all aspect. There are few gaps which have to be addressed.
1. Update the Reference for Cost of Wound Management
2. Clarify if analysis has focused PRP Treatment for all types of Chronic Wound Management or just 2 or 3 types of Wounds which authors have listed in Table 1?? If authors are focusing only specific wounds then it is suggested to modify the title.
Reviewer 2 Report
1. The main question addressed by the research is to find out the therapeutic effect of PRP on chronic wound
2. There is no showing good novelty compared with other similar publications.
3. The methodology is well structured, but there is no part to evaluate risk of bias.
4. The conclusions are consistent with the evidence and arguments presented
5. The references are appropriate.
6. There is no part to evaluate risk of bias on the figures.
Reviewer 3 Report
This manuscript is based on systematic review and meta-analysis of RCT on patients with chronic wounds treated with platelet-rich plasma. The materials, methods and results are well described and presented. The paper addresses an important and interesting aspect of current issue in wound healing. The parameters discussed (complete wound, closure, reduction of wound area, healing time, infection rate and pain) are the standard parameters assessed in wound healing research. However, as this review and the trials are based on clinical aspects without taking in account of cost and other non-clinical factors, it is more appropriate to use the term “clinical efficacy” rather than ‘efficacy’ to give better context to the review.
Round 2
Reviewer 2 Report
- It is not necessary to revise manuscript.